# Computing Kantorovich-Wasserstein Distances on $d$-dimensional histograms using $(d + 1)$-partite graphs

**Gennaro Auricchio, Stefano Gualandi, Marco Veneroni**
Università degli Studi di Pavia, Dipartimento di Matematica "F. Casorati"
`gennaro.auricchio01@universitadipavia.it,`
`stefano.gualandi@unipv.it, marco.veneroni@unipv.it`

**Federico Bassetti**
Politecnico di Milano, Dipartimento di Matematica
`federico.bassetti@polimi.it`

## Abstract

This paper presents a novel method to compute the exact Kantorovich-Wasserstein distance between a pair of $d$-dimensional histograms having $n$ bins each. We prove that this problem is equivalent to an uncapacitated minimum cost flow problem on a $(d + 1)$-partite graph with $(d + 1)n$ nodes and $dn^{\frac{d+1}{d}}$ arcs, whenever the cost is separable along the principal $d$-dimensional directions. We show numerically the benefits of our approach by computing the Kantorovich-Wasserstein distance of order 2 among two sets of instances: gray scale images and $d$-dimensional bio medical histograms. On these types of instances, our approach is competitive with state-of-the-art optimal transport algorithms.

## 1 Introduction

The computation of a measure of similarity (or dissimilarity) between pairs of objects is a crucial subproblem in several applications in Computer Vision [24, 25, 22], Computational Statistic [17], Probability [6, 8], and Machine Learning [29, 12, 14, 5]. In mathematical terms, in order to compute the similarity between a pair of objects, we want to compute a *distance*. If the distance is equal to zero the two objects are considered to be equal; the more the two objects are different, the greater is their distance value. For instance, the Euclidean norm is the most used distance function to compare a pair of points in $\mathbb{R}^d$. Note that the Euclidean distance requires only $O(d)$ operations to be computed. When computing the distance between complex discrete objects, such as for instance a pair of discrete measures, a pair of images, a pair of $d$-dimensional histograms, or a pair of clouds of points, the Kantorovich-Wasserstein distance [31, 30] has proved to be a relevant distance function [24], which has both nice mathematical properties and useful practical implications. Unfortunately, computing the Kantorovich-Wasserstein distance requires the solution of an optimization problem. Even if the optimization problem is polynomially solvable, the size of practical instances to be solved is very large, and hence the computation of Kantorovich-Wasserstein distances implies an important computational burden.

The optimization problem that yields the Kantorovich-Wasserstein distance can be solved with different methods. Nowadays, the most popular methods are based on (i) the Sinkhorn's algorithm [11, 28, 3], which solves (heuristically) a regularized version of the basic optimal transport problem, and (ii) Linear Programming-based algorithms [13, 15, 20], which exactly solve the basic optimal transport problem by formulating and solving an equivalent uncapacitated minimum cost flow problem. For a nice overview of both computational approaches, we refer the reader to Chapters 2 and 3 in [23], and the references therein contained.

In this paper, we propose a Linear Programming-based method to speed up the computation of Kantorovich-Wasserstein distances of order 2, which exploits the structure of the ground distance to formulate an uncapacitated minimum cost flow problem. The flow problem is then solved with a state-of-the-art implementation of the well-known Network Simplex algorithm [16].

Our approach is along the line of research initiated in [19], where the authors proposed a very efficient method to compute Kantorovich-Wasserstein distances of order 1 (i.e., the so–called *Earth Mover Distance*), whenever the ground distance between a pair of points is the $\ell_1$ norm. In [19], the structure of the $\ell_1$ ground distance and of regular $d$-dimensional histograms is exploited to define a very small flow network. More recently, this approach has been successfully generalized in [7] to the case of $\ell_\infty$ and $\ell_2$ norms, providing both exact and approximations algorithms, which are able to compute distances between pairs of $512 \times 512$ gray scale images. The idea of speeding up the computation of Kantorovich-Wasserstein distances by defining a minimum cost flow on smaller structured flow networks is also used in [22], where a truncated distance is used as ground distance in place of a $\ell_p$ norm.

The outline of this paper is as follows. Section 2 reviews the basic notion of discrete optimal transport and fixes the notation. Section 3 contains our main contribution, that is, Theorem 1 and Corollary 2, which permits to speed-up the computation of Kantorovich-Wasserstein distances of order 2 under quite general assumptions. Section 4 presents numerical results of our approaches, compared with the Sinkhorn's algorithm as implemented in [11] and a standard Linear Programming formulation on a complete bipartite graph [24]. Finally, Section 5 concludes the paper.

## 2 Discrete Optimal Transport: an Overview

Let $X$ and $Y$ be two discrete spaces. Given two probability vectors $\mu$ and $\nu$ defined on $X$ and $Y$, respectively, and a cost $c : X \times Y \to \mathbb{R}_+$, the *Kantorovich-Rubinshtein functional* between $\mu$ and $\nu$ is defined as

$$\mathcal{W}_c(\mu, \nu) = \inf_{\pi \in \Pi(\mu,\nu)} \sum_{(x,y) \in X \times Y} c(x,y)\pi(x,y) \tag{1}$$

where $\Pi(\mu, \nu)$ is the set of all the probability measures on $X \times Y$ with marginals $\mu$ and $\nu$, i.e. the probability measures $\pi$ such that $\sum_{y \in Y} \pi(x,y) = \mu(x)$ and $\sum_{x \in X} \pi(x,y) = \nu(y)$, for every $(x,y)$ in $X \times Y$. Such probability measures are sometimes called transport plans or couplings for $\mu$ and $\nu$. An important special case is when $X = Y$ and the cost function $c$ is a distance on $X$. In this case $\mathcal{W}_c$ is a distance on the simplex of probability vectors on $X$, also known as *Kantorovich-Wasserstein distance* of order 1.

We remark that **the Kantorovich-Wasserstein distance of order** $p$ can be defined, more in general, for arbitrary probability measures on a metric space $(X, \delta)$ by

$$W_p(\mu, \nu) := \left( \inf_{\pi \in \Pi(\mu,\nu)} \int_{X \times X} \delta^p(x,y)\pi(dxdy) \right)^{\min(1/p,1)} \tag{2}$$

where now $\Pi(\mu, \nu)$ is the set of all probability measures on the Borel sets of $X \times X$ that have marginals $\mu$ and $\nu$, see, e.g., [4]. The infimum in (2) is attained, and any probability $\pi$ which realizes the minimum is called an *optimal transport plan*.

The Kantorovich-Rubinshtein transport problem in the discrete setting can be seen as a special case of the following Linear Programming problem, where we assume now that $\mu$ and $\nu$ are generic vectors of dimension $n$, with positive components,

$$(P) \quad \min \quad \sum_{x \in X} \sum_{y \in Y} c(x,y)\pi(x,y) \tag{3}$$

$$\text{s.t.} \quad \sum_{y \in Y} \pi(x,y) \leq \mu(x) \qquad\qquad \forall x \in X \tag{4}$$

$$\sum_{x \in X} \pi(x,y) \geq \nu(y) \qquad\qquad \forall y \in Y \tag{5}$$

$$\pi(x,y) \geq 0. \tag{6}$$

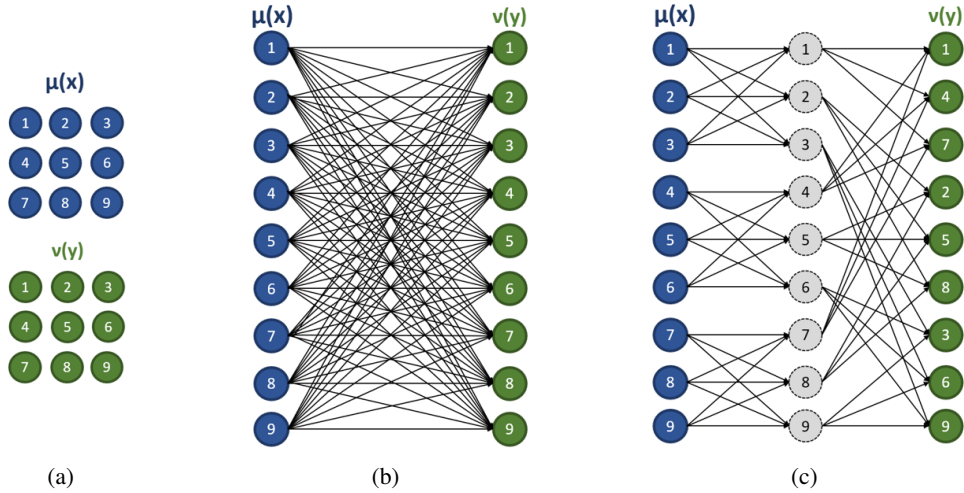

Figure 1: (a) Two given 2-dimensional histograms of size $N \times N$, with $N = 3$; (b) Complete bipartite graph with $N^4$ arcs; (c): 3-partite graph with $(d+1)N^3$ arcs.

If $\sum_x \mu(x) = \sum_y \nu(y)$ we have the so-called *balanced* transportation problem, otherwise the transportation problem is said to be *unbalanced* [18, 10]. For balanced optimal transport problems, constraints (4) and (5) must be satisfied with equality, and the problem reduces to the Kantorovich transport problem (up to normalization of the vectors $\mu$ and $\nu$).

Problem (P) is related to the so-called *Earth Mover's distance*. In this case, $X, Y \subset \mathbb{R}^d$, $x$ and $y$ are the centers of two data clusters, and $\mu(x)$ and $\nu(y)$ give the number of points in the respective cluster. Finally, $c(x, y)$ is some measure of dissimilarity between the two clusters $x$ and $y$. Once the optimal transport $\pi^*$ is determined, the Earth Mover's distance between $\mu$ and $\nu$ is defined as (e.g., see [24])

$$EMD(\mu, \nu) = \frac{\sum_{x \in X} \sum_{y \in Y} c(x, y) \pi^*(x, y)}{\sum_{x \in X} \sum_{y \in Y} \pi^*(x, y)}.$$

Problem (P) can be formulated as an uncapacitated minimum cost flow problem on a bipartite graph defined as follows [2]. The bipartite graph has two partitions of nodes: the first partition has a node for each point $x$ of $X$, and the second partition has a node for each point $y$ of $Y$. Each node $x$ of the first partition has a supply of mass equal to $\mu(x)$, each node of the second partition has a demand of $\nu(y)$ units of mass. The bipartite graph has an (uncapacitated) arc for each element in the Cartesian product $X \times Y$ having cost equal to $c(x, y)$. The minimum cost flow problem defined on this graph yields the optimal transport plan $\pi^*(x, y)$, which indeed is an optimal solution of problem (3)–(6). For instance, in case of a regular 2D dimensional histogram of size $N \times N$, that is, having $n = N^2$ bins, we get a bipartite graph with $2N^2$ nodes and $N^4$ arcs (or $2n$ nodes and $n^2$ arcs). Figure 1–(a) shows an example for a $3 \times 3$ histogram, and Figure 1–(b) gives the corresponding complete bipartite graph.

In this paper, we focus on the case $p = 2$ in equation (2) and the ground distance function $\delta$ is the Euclidean norm $\ell_2$, that is the Kantorovich-Wasserstein distance of order 2, which is denoted by $W_2$. We provide, in the next section, an equivalent formulation on a smaller $(d+1)$-partite graph.

## 3   Formulation on $(d+1)$-partite Graphs

For the sake of clarity, but without loss of generality, we present first our construction considering 2-dimensional histograms and the $\ell_2$ Euclidean ground distance. Then, we discuss how our construction can be generalized to any pair of $d$-dimensional histograms.

Let us consider the following flow problem: let $\mu$ and $\nu$ be two probability measures over a $N \times N$ regular grid denoted by $G$. In the following paragraphs, we use the notation sketched in Figure 2. In addition, we define the set $U := \{1, \ldots, N\}$.

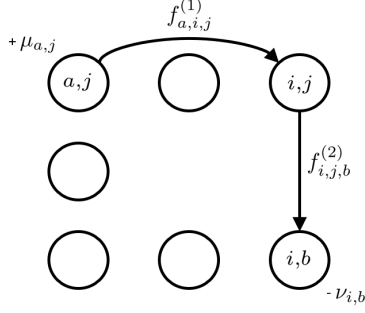

Figure 2: Basic notation used in Section 3: in order to send a unit of flow from point $(a, j)$ to point $(i, b)$, we either send a unit of flow directly along arc $((a, j), (i, b))$ of cost $c((a, j), (i, b)) = (a - i)^2 + (j - b)^2$, or, we first send a unit of flow from $(a, j)$ to $(i, j)$, and then from $(i, j)$ to $(i, b)$, having total cost $c((a, j), (i, j)) + c((i, j), (i, b)) = (a-i)^2+(j-j)^2+(i-i)^2+(j-b)^2 = (a - i)^2 + (j - b)^2 = c((a, j), (i, b))$. Indeed, the cost of the two different path is exactly the same.

Since we are considering the $\ell_2$ norm as ground distance, we minimize the functional

$$R : (F_1, F_2) \to \sum_{i,j=1}^{N} \left[ \sum_{a=1}^{N} (a - i)^2 f_{a,i,j}^{(1)} + \sum_{b=1}^{N} (j - b)^2 f_{i,j,b}^{(2)} \right] \tag{7}$$

among all $F_i = \{f_{a,b,c}^{(i)}\}$, with $a, b, c \in \{1, ..., N\}$ real numbers (i.e., flow variables) satisfying the following constraints

$$\sum_{i=1}^{N} f_{a,i,j}^{(1)} = \mu_{a,j}, \qquad \forall a, j \in U \times U \tag{8}$$

$$\sum_{j=1}^{N} f_{i,j,b}^{(2)} = \nu_{i,b}, \qquad \forall i, b \in U \times U \tag{9}$$

$$\sum_{a} f_{a,i,j}^{(1)} = \sum_{b} f_{i,j,b}^{(2)}, \qquad \forall i, j \in U \times U, a \in U, b \in U. \tag{10}$$

Constraints (8) impose that the mass $\mu_{a,j}$ at the point $(a, j)$ is moved to the points $(k, j)_{k=1,...,N}$. Constraints (9) force the point $(i, b)$ to receive from the points $(i, l)_{l=1,...,N}$ a total mass of $\nu_{i,b}$. Constraints (10) require that all the mass that goes from the points $(a, j)_{a=1,...,N}$ to the point $(i, j)$ is moved to the points $(i, b)_{b=1,...,N}$. We call a pair $(F_1, F_2)$ satisfying the constraints (8)–(10) a *feasible flow* between $\mu$ and $\nu$. We denote by $\mathcal{F}(\mu, \nu)$ the set of all feasible flows between $\mu$ and $\nu$.

Indeed, we can formulate the minimization problem defined by (7)–(10) as an uncapacitated minimum cost flow problem on a tripartite graph $T = (V, A)$. The set of nodes of $T$ is $V := V^{(1)} \cup V^{(2)} \cup V^{(3)}$, where $V^{(1)}, V^{(2)}$ and $V^{(3)}$ are the nodes corresponding to three $N \times N$ regular grids. We denote by $(i, j)^{(l)}$ the node of coordinates $(i, j)$ in the grid $V^{(l)}$. We define the two disjoint set of arcs between the successive pairs of node partitions as

$$A^{(1)} := \{((a, j)^{(1)}, (i, j)^{(2)}) \mid i, a, j \in U\}, \tag{11}$$

$$A^{(2)} := \{((i, j)^{(2)}, (i, b)^{(3)}) \mid i, b, j \in U\}, \tag{12}$$

and, hence, the arcs of $T$ are $A := A^{(1)} \cup A^{(2)}$. Note that in this case the graph $T$ has $3N^2$ nodes and $2N^3$ arcs. Whenever $(F_1, F_2)$ is a feasible flow between $\mu$ and $\nu$, we can think of the values $f_{a,i,j}^{(1)}$ as the quantity of mass that travels from $(a, j)$ to $(i, j)$ or, equivalently, that moves along the arc $((a, j), (i, j))$ of the tripartite graph, while the values $f_{i,j,b}^{(2)}$ are the mass moving along the arc $((i, j), (i, b))$ (e.g., see Figures 1–(c) and 2).

Now we can give an idea of the roles of the sets $V^{(1)}, V^{(2)}$ and $V^{(3)}$: $V^{(1)}$ is the node set where is drawn the initial distribution $\mu$, while on $V^{(3)}$ it is drawn the final configuration of the mass $\nu$. The node set $V^{(2)}$ is an auxiliary grid that hosts an intermediate configuration between $\mu$ and $\nu$.

We are now ready to state our main contribution.

**Theorem 1.** *For each measure $\pi$ on $G \times G$ that transports $\mu$ into $\nu$, we can find a feasible flow $(F_1, F_2)$ such that*

$$R(F_1, F_2) = \sum_{((a,j),(i,b))} ((a - i)^2 + (b - j)^2) \pi_{(a,j),(i,b)}. \tag{13}$$

*Proof. (Sketch).* We will only show how to build a feasible flow starting from a transport plan, the inverse building uses a more technical lemma (the so–called *gluing lemma* [4, 31]) and can be found in the Additional Material. Let $\pi$ be a transport plan, if we write explicitly the ground distance $\ell_2((a,j),(i,b))$ we find that

$$\sum_{((a,j),(i,b))} \ell_2((a,j),(i,b))\pi_{((a,j),(i,b))} = \sum_{((a,j),(i,b))} ((a-i)^2 + (j-b)^2)\pi_{((a,j),(i,b))}$$

$$= \sum_{j,i} \left[ \sum_{a,b}(a-i)^2\pi_{((a,j),(i,b))} + \sum_{a,b}(j-b)^2\pi_{((a,j),(i,b))} \right].$$

If we set $f^{(1)}_{a,i,j} = \sum_b \pi_{((a,j),(i,b))}$ and $f^{(2)}_{i,j,b} = \sum_a \pi_{((a,j),(i,b))}$ we find

$$\sum_{((a,j),(i,b))} \ell_2((a,j),(i,b))\pi_{((a,j),(i,b))} = \sum_{i,j}^n \left[ \sum_a^n(a-i)^2 f^{(1)}_{a,i,j} + \sum_b^n(j-b)^2 f^{(2)}_{i,j,b} \right].$$

In order to conclude we have to prove that those $f^{(1)}_{a,i,j}$ and $f^{(2)}_{i,j,b}$ satisfy the constraints (8)–(10).

By definition we have

$$\sum_i f^{(1)}_{a,i,j} = \sum_i \sum_b \pi_{((a,j),(i,b))} = \mu_{a,j},$$

thus proving (8); similarly, it is possible to check constraint (9). The constraint (10) also follows easily since

$$\sum_a f^{(1)}_{a,i,j} = \sum_a \sum_b \pi_{((a,j),(i,b))} = \sum_b f^{(2)}_{i,j,b}.$$

$\square$

As a straightforward, yet fundamental, consequence we have the following result.

**Corollary 1.** *If we set $c((a,j),(i,b)) = (a-i)^2 + (j-b)^2$ then, for any discrete measures $\mu$ and $\nu$, we have that*

$$W_2^2(\mu,\nu) = \min_{\mathcal{F}(\mu,\nu)} R(F_1, F_2). \tag{14}$$

Indeed, we can compute the Kantorovich-Wasserstein distance of order 2 between a pair of discrete measures $\mu, \nu$, by solving an uncapacitated minimum cost flow problem on the given tripartite graph $T := (V^{(1)} \cup V^{(2)} \cup V^{(3)}, A^{(1)} \cup A^{(2)})$.

We remark that our approach is very general and it can be directly extended to deal with the following generalizations.

**More general cost functions.** The structure that we have exploited of the Euclidean distance $\ell_2$ is present in any cost function $c : G \times G \to [0, \infty]$ that is separable, i.e., has the form

$$c(x,y) = c^{(1)}(x_1, y_1) + c^{(2)}(x_2, y_2),$$

where both $c^{(1)}$ and $c^{(2)}$ are positive real valued functions defined over $G$. We remark that the whole class of costs $c_p(x,y) = (x_1 - y_1)^p + (x_2 - y_2)^p$ is of that kind, so we can compute any of the Kantorovich-Wasserstein distances related to each $c_p$.

**Higher dimensional grids.** Our approach can handle discrete measures in spaces of any dimension $d$, that is, for instance, any $d$-dimensional histogram. In dimension $d = 2$, we get a tripartite graph because we decomposed the transport along the two main directions. If we have a problem in dimension $d$, we need a $(d+1)$-plet of grids connected by arcs oriented as the $d$ fundamental directions, yielding a $(d+1)$-partite graph. As the dimension $d$ grows, our approach gets faster and more memory efficient than the standard formulation given on a bipartite graph.

In the Additional Material, we present a generalization of Theorem 1 to any dimension $d$ and to *separable* cost functions $c(x,y)$.

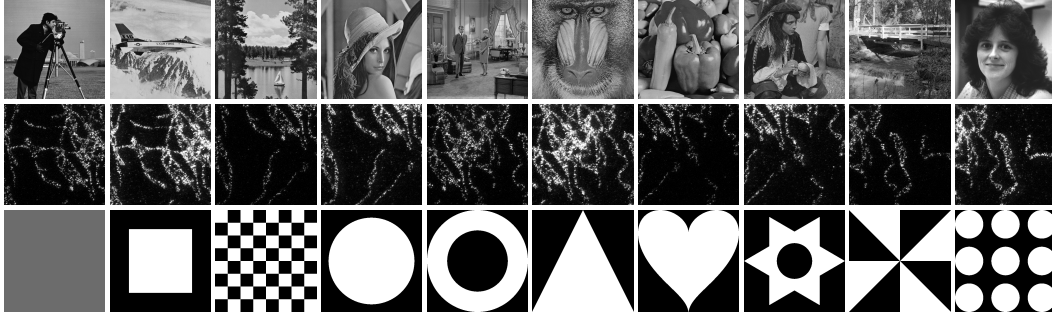

Figure 3: DOTmark benchmark: Classic, Microscopy, and Shapes images.

## 4 Computational Results

In this section, we report the results obtained on two different set of instances. The goal of our experiments is to show how our approach scales with the size of the histogram $N$ and with the dimension of the histogram $d$. As cost distance $c(x, y)$, with $x, y \in \mathbb{R}^d$, we use the squared $\ell_2$ norm. As problem instances, we use the gray scale images (i.e., 2-dimensional histograms) proposed by the DOTMark benchmark [26], and a set of $d$-dimensional histograms obtained by bio medical data measured by flow cytometer [9].

**Implementation details.** We run our experiments using the Network Simplex as implemented in the Lemon C++ graph library[1], since it provides the fastest implementation of the Network Simplex algorithm to solve uncapacitated minimum cost flow problems [16]. We did try other state-of-the-art implementations of combinatorial algorithm for solving min cost flow problems, but the Network Simplex of the Lemon graph library was the fastest by a large margin. The tests are executed on a gaming laptop with Windows 10 (64 bit), equipped with an Intel i7-6700HQ CPU and 16 GB of Ram. The code was compiled with MS Visual Studio 2017, using the ANSI standard C++17. The code execution is single threaded. The Matlab implementation of the Sinkhorn's algorithm [11] runs in parallel on the CPU cores, but we do not use any GPU in our test. The C++ and Matlab code we used for this paper is freely available at `http://stegua.github.io/dpartion-nips2018`.

**Results for the DOTmark benchmark.** The DOTmark benchmark contains 10 classes of gray scale images related to randomly generated images, classical images, and real data from microscopy images of mitochondria [26]. In each class there are 10 different images. Every image is given in the data set at the following pixel resolutions: $32 \times 32$, $64 \times 64$, $128 \times 128$, $256 \times 256$, and $512 \times 512$. The images in Figure 3 are respectively the *ClassicImages*, *Microscopy*, and *Shapes* images (one class for each row), shown at highest resolution.

In our test, we first compared five approaches to compute the Kantorovich-Wasserstein distances on images of size $32 \times 32$:

1. **EMD**: The implementation of Transportation Simplex provided by [24], known in the literature as EMD code, that is an exact general method to solve optimal transport problem. We used the implementation in the programming language C, as provided by the authors, and compiled with all the compiler optimization flags active.

2. **Sinkhorn**: The Matlab implementation of the Sinkhorn's algorithm[2] [11], that is an approximate approach whose performance in terms of speed and numerical accuracy depends on a parameter $\lambda$: for smaller values of $\lambda$, the algorithm is faster, but the solution value has a large gap with respect to the optimal value of the transportation problem; for larger values of $\lambda$, the algorithm is more accurate (i.e., smaller gap), but it becomes slower. Unfortunately, for very large value of $\lambda$ the method becomes numerically unstable. The best value of $\lambda$ is very problem dependent. In our tests, we used $\lambda = 1$ and $\lambda = 1.5$. The second value,

$\lambda = 1.5$, is the largest value we found for which the algorithm computes the distances for all the instances considered without facing numerical issues.

3. **Improved Sinkhorn**: We implemented in Matlab an improved version of the Sinkhorn's algorithm, specialized to compute distances over regular 2-dimensional grids [28, 27]. The main idea is to improve the matrix-vector operations that are the true computational bottleneck of Sinkhorn's algorithm, by exploiting the structure of the cost matrix. Indeed, there is a parallelism with our approach to the method presented in [28], since both exploits the geometric cost structure. In [28], the authors proposes a general method that exploits a heat kernel to speed up the matrix-vector products. When the discrete measures are defined over a regular 2-dimensional grid, the cost matrix used by the Sinkhorn's algorithm can be obtained using a Kronecker product of two smaller matrices. Hence, instead of performing a matrix-vector product using a matrix of dimension $N \times N$, we perform two matrix-matrix products over matrices of dimension $\sqrt{N} \times \sqrt{N}$, yielding a significant runtime improvement. In addition, since the smaller matrices are Toeplitz matrices, they can be embedded into circulant matrices, and, as consequence, it is possible to employ a Fast Fourier Transform approach to further speed up the computation. Unfortunately, the Fast Fourier Transform makes the approach still more numerical unstable, and we did not used it in our final implementation.

4. **Bipartite**: The bipartite formulation presented in Figure 1–(b), which is the same as [24], but it is solved with the Network Simplex implemented in the Lemon Graph library [16].

5. 3-**partite**: The 3-partite formulation proposed in this paper, which for 2-dimensional histograms is represented in 1–(c). Again, we use the Network Simplex of the Lemon Graph Library to solve the corresponding uncapacitated minimum cost flow problem.

Tables 1(a) and 1(b) report the averages of our computational results over different classes of images of the DOTMark benchmark. Each class of gray scale image contains 10 instances, and we compute the distance between every possible pair of images within the same class: the first image plays the role of the source distribution $\mu$, and the second image gives the target distribution $\nu$. Considering all pairs within a class, it gives 45 instances for each class. We report the means and the standard deviations (between brackets) of the runtime, measured in seconds. Table 1(a) shows in the second column the runtime for EMD [24]. The third and fourth columns gives the runtime and the optimality gap for the Sinkhorn's algorithm with $\lambda = 1$; the 6-$th$ and 7-$th$ columns for $\lambda = 1.5$. The percentage gap is computed as $\text{Gap} = \frac{UB - opt}{opt} \cdot 100$, where $UB$ is the upper bound computed by the Sinkhorn's algorithm, and $opt$ is the optimal value computed by EMD. The last two columns report the runtime for the bipartite and 3-partite approaches presented in this paper.

Table 1(b) compares our 3-partite formulation with the Improved Sinkhorn's algorithm [28, 27], reporting the same statistics of the previous table. In this case, we run the Improved Sinkhorn using three values of the parameter $\lambda$, that are, 1.0, 1.25, and 1.5. While the Improved Sinkhorn is indeed much faster that the general algorithm as presented in [11], it does suffer of the same numerical stability issues, and, it can yield very poor percentage gap to the optimal solution, as it happens for the GRFrough and the WhiteNoise classes, where the optimality gaps are on average 31.0% and 39.2%, respectively.

As shown in Tables 1(a) and 1(b), the 3-partite approach is clearly faster than any of the alternatives considered here, despite being an exact method. In addition, we remark that, even on the bipartite formulation, the Network Simplex implementation of the Lemon Graph library is order of magnitude faster than EMD, and hence it should be the best choice in this particular type of instances. We remark that it might be unfair to compare an algorithm implemented in C++ with an algorithm implemented in Matlab, but still, the true comparison is on the solution quality more than on the runtime. Moreover, when implemented on modern GPU that can fully exploit parallel matrix-vector operations, the Sinkhorn's algorithm can run much faster, but they cannot improve the optimality gap.

In order to evaluate how our approach scale with the size of the images, we run additional tests using images of size $64 \times 64$ and $128 \times 128$. Table 2 reports the results for the bipartite and 3-partite approaches for increasing size of the 2-dimensional histograms. The table report for each of the two approaches, the number of vertices $|V|$ and of arcs $|A|$, and the means and standard deviations of the runtime. As before, each row gives the averages over 45 instances. Table 2 shows that the 3-partite approach is clearly better (i) in terms of memory, since the 3-partite graph has a fraction of the number of arcs, and (ii) of runtime, since it is at least an order of magnitude faster in computation

|  | EMD [24] | Sinkhorn [11] | | | | Bipartite | 3-partite |
|  |  | $\lambda = 1$ | | $\lambda = 1.5$ | | | |
| Image Class | Runtime | Runtime | Gap | Runtime | Gap | Runtime | Runtime |
|---|---|---|---|---|---|---|---|
| Classic | 24.0 (3.3) | 6.0 (0.5) | 17.3% | 8.9 (0.7) | 9.1% | 0.54 (0.05) | 0.07 (0.01) |
| Microscopy | 35.0 (3.3) | 3.5 (1.0) | 2.4% | 5.3 (1.4) | 1.2% | 0.55 (0.03) | 0.08 (0.01) |
| Shapes | 25.2 (5.3) | 1.6 (1.1) | 5.6% | 2.5 (1.6) | 3.0% | 0.50 (0.07) | 0.05 (0.01) |

(a)

|  | Improved Sinkhorn [28, 27] | | | | | | 3-partite |
|  | $\lambda = 1$ | | $\lambda = 1.25$ | | $\lambda = 1.5$ | | |
| Image Class | Runtime | Gap | Runtime | Gap | Runtime | Gap | Runtime |
|---|---|---|---|---|---|---|---|
| CauchyDensity | 0.22 (0.15) | 2.8% | 0.33 (0.23) | 2.0% | 0.41 (0.28) | 1.5% | 0.07 (0.01) |
| Classic | 0.20 (0.01) | 17.3% | 0.31 (0.02) | 12.4% | 0.39 (0.03) | 9.1% | 0.07 (0.01) |
| GRFmoderate | 0.19 (0.01) | 12.6% | 0.29 (0.02) | 9.0% | 0.37 (0.03) | 6.6% | 0.07 (0.01) |
| GRFrough | 0.19 (0.01) | 58.7% | 0.29 (0.01) | 42.1% | 0.38 (0.02) | 31.0% | 0.05 (0.01) |
| GRFsmooth | 0.20 (0.02) | 4.3% | 0.30 (0.04) | 3.1% | 0.38 (0.04) | 2.2% | 0.08 (0.01) |
| LogGRF | 0.22 (0.05) | 1.3% | 0.32 (0.08) | 0.9% | 0.40 (0.13) | 0.7% | 0.08 (0.01) |
| LogitGRF | 0.22 (0.02) | 4.7% | 0.33 (0.03) | 3.3% | 0.42 (0.04) | 2.5% | 0.07 (0.02) |
| Microscopy | 0.18 (0.03) | 2.4% | 0.27 (0.04) | 1.7% | 0.34 (0.05) | 1.2% | 0.08 (0.02) |
| Shapes | 0.11 (0.04) | 5.6% | 0.16 (0.06) | 4.0% | 0.20 (0.07) | 3.0% | 0.05 (0.01) |
| WhiteNoise | 0.18 (0.01) | 76.3% | 0.28 (0.01) | 53.8% | 0.37 (0.02) | 39.2% | 0.04 (0.00) |

(b)

Table 1: Comparison of different approaches on $32 \times 32$ images. The runtime (in seconds) is given as "Mean (StdDev)". The gap to the optimal value *opt* is computed as $\frac{UB-opt}{opt} \cdot 100$, where $UB$ is the upper bound computed by Sinkhorn's algorithm. Each row reports the averages over 45 instances.

|  |  | Bipartite | | | 3-partite | | |
| Size | Image Class | $|V|$ | $|A|$ | Runtime | $|V|$ | $|A|$ | Runtime |
|---|---|---|---|---|---|---|---|
| $64 \times 64$ | Classic | 8 193 | 16 777 216 | 16.3 (3.6) | 12 288 | 524 288 | 2.2 (0.2) |
|  | Microscopy |  |  | 11.7 (1.4) |  |  | 1.0 (0.2) |
|  | Shape |  |  | 13.0 (3.9) |  |  | 1.1 (0.3) |
| $128 \times 128$ | Classic | 32 768 | 268 435 456 | 1 368 (545) | 49 152 | 4 194 304 | 36.2 (5.4) |
|  | Microscopy |  |  | 959 (181) |  |  | 23.0 (4.8) |
|  | Shape |  |  | 983 (230) |  |  | 17.8 (5.2) |

Table 2: Comparison of the bipartite and the 3-partite approaches on 2-dimensional histograms.

time. Indeed, the 3-partite formulation is better essentially because it exploits the structure of the ground distance $c(x, y)$ used, that is, the squared $\ell_2$ norm.

**Flow Cytometry biomedical data.** Flow cytometry is a laser-based biophysical technology used to study human health disorders. Flow cytometry experiments produce huge set of data, which are very hard to analyze with standard statistics methods and algorithms [9]. Currently, such data is used to study the correlations of only two factors (e.g., biomarkers) at the time, by visualizing 2-dimensional histograms and by measuring the (dis-)similarity between pairs of histograms [21]. However, during a flow cytometry experiment up to hundreds of factors (biomarkers) are measured and stored in digital format. Hence, we can use such data to build $d$-dimensional histograms that consider up to $d$ biomarkers at the time, and then comparing the similarity among different individuals by measuring the distance between the corresponding histograms. In this work, we used the flow cytometry data related to *Acute Myeloid Leukemia (AML)*, available at `http://flowrepository.`

|  |  |  |  | Bipartite Graph |  | $(d+1)$-partite Graph |  |  |
| N | $d$ | $n$ | $\|V\|$ | $\|A\|$ | Runtime | $\|V\|$ | $\|A\|$ | Runtime |
|---|---|---|---|---|---|---|---|---|
| 16 | 2 | 256 | 512 | 65 536 | 0.024 (0.01) | 768 | 8 192 | 0.003 (0.00) |
|  | 3 | 4 096 | 8 192 | 16 777 216 | 38.2 (14.0) | 16 384 | 196 608 | 0.12 (0.02) |
|  | 4 | 65 536 |  | *out-of-memory* |  | 327 680 | 4 194 304 | 4.8 (0.84) |
| 32 | 2 | 1 024 | 2 048 | 1 048 756 | 0.71 (0.14) | 3072 | 65 536 | 0.04 (0.01) |
|  | 3 | 32 768 |  | *out-of-memory* |  | 131 072 | 3 145 728 | 5.23 (0.69) |

Table 3: Comparison between the bipartite and the $(d+1)$-partite approaches on Flow Cytometry data.

`org/id/FR-FCM-ZZYA`, which contains cytometry data for 359 patients, classified as "normal" or affected by AML. This dataset has been used by the bioinformatics community to run clustering algorithms, which should predict whether a new patient is affected by AML [1].

Table 3 reports the results of computing the distance between pairs of $d$-dimensional histograms, with $d$ ranging in the set $\{2, 3, 4\}$, obtained using the AML biomedical data. Again, the first $d$-dimensional histogram plays the role of the source distribution $\mu$, while the second histogram gives the target distribution $\nu$. For simplicity, we considered regular histograms of size $n = N^d$ (i.e., $n$ is the total number of bins), using $N = 16$ and $N = 32$. Table 3 compares the results obtained by the bipartite and $(d+1)$-partite approach, in terms of graph size and runtime. Again, the $(d+1)$-partite approach, by exploiting the structure of the ground distance, outperforms the standard formulation of the optimal transport problem. We remark that for $N = 32$ and $d = 3$, we pass for going out-of-memory with the bipartite formulation, to compute the distance in around 5 seconds with the 4-partite formulation.

## 5   Conclusions

In this paper, we have presented a new network flow formulation on $(d+1)$-partite graphs that can speed up the optimal solution of transportation problems whenever the ground cost function $c(x, y)$ (see objective function (3)) has a separable structure along the main $d$ directions, such as, for instance, the squared $\ell_2$ norm used in the computation of the Kantorovich-Wasserstein distance of order 2.

Our computational results on two different datasets show how our approach scales with the size of the histograms $N$ and with the dimension of the histograms $d$. Indeed, by exploiting the cost structure, the proposed approach is better in term of memory consumption, since it has only $dn^{\frac{d+1}{d}}$ arcs instead of $n^2$. In addition, it is much faster since it has to solve an uncapacitated minimum cost flow problem on a much smaller flow network.

**Acknowledgments**

We are deeply indebted to Giuseppe Savaré, for introducing us to optimal transport and for many stimulating discussions and suggestions. We thanks Mattia Tani for a useful discussion concerning the Improved Sinkhorn's algorithm.

This research was partially supported by the Italian Ministry of Education, University and Research (MIUR): Dipartimenti di Eccellenza Program (2018–2022) - Dept. of Mathematics "F. Casorati", University of Pavia.

The last author's research is partially supported by "PRIN 2015. 2015SNS29B-002. Modern Bayesian nonparametric methods".

## Footnotes

[1] `http://lemon.cs.elte.hu` (last visited on October, 26th, 2018)

[2] `http://marcocuturi.net/SI.html` (last visited on October, 26th, 2018)

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
