[Supplementary Material]

## Additional Material

**Lemma 2.** *(Gluing Lemma, [4, 31]) Let $\pi^{(1)}$ and $\pi^{(2)}$ be two discrete probability measures in $\mathbb{R}^d \times \mathbb{R}^d$ such that*

$$\sum_{(b_1,...,b_d)} \pi^{(1)}(a_1,...,a_d; b_1,...,b_d) = \sum_{(b_1,...,b_d)} \pi^{(2)}(b_1,...,b_d; c_1,...,c_d)$$

*Then there exists a discrete probability measure $\pi$ on $\mathbb{R}^d \times \mathbb{R}^d \times \mathbb{R}^d$ such that*

$$\sum_{(c_1,...,c_d)} \pi(a_1,...,a_d; b_1,...,b_d; c_1,...,c_d) = \pi^{(1)}(a_1,...,a_d; b_1,...,b_d)$$

*and*

$$\sum_{(a_1,...,a_d)} \pi(a_1,...,a_d; b_1,...,b_d; c_1,...,c_d) = \pi^{(2)}(b_1,...,b_d; c_1,...,c_d).$$

Let us take $\mu$, $\nu$ two probability measures and a ground distance of the form

$$c((a_1,...,a_d),(b_1,...,b_d)) = \sum_{i=1}^{d} \Delta_i(a_i, b_i). \tag{15}$$

We can then define

$$R(F_1,...,F_d) = \sum_{i=1}^{d} \left[ \sum_{b_1,...,b_{i-1},a_i,...,a_d,b_i} \Delta_i(a_i,b_i) f^{(i)}_{b_1,...,b_{i-1},a_i,...,a_d,b_i} \right], \tag{16}$$

where

$$F_i = \{ f^{(i)}_{b_1,...,b_{i-1},a_i,...,a_d,b_i} \}$$

are a $N^{d+1}$-plet of real values satisfying the two congruence conditions

$$\sum_{b_1} f^{(1)}_{a_1,...,a_d,b_1} = \mu(a_1,...,a_d), \tag{17}$$

$$\sum_{a_N} f^{(d)}_{b_1,...,b_{d-1},a_d,b_d} = \nu(b_1,...,b_d) \tag{18}$$

and the following $d-1$ connection conditions

$$\sum_{a_i} f^{(i)}_{b_1,...,b_{i-1},a_i,...,a_d,b_i} = \sum_{b_{i+1}} f^{(i+1)}_{b_1,...,b_i,a_{i+1},...,a_d,b_{i+1}} \tag{19}$$

for $i = 1,..,d-1$. We will call the $d-$plet of $(F_1,...,F_d)$ a flow chart between $\mu$ and $\nu$.

The set of all possible flow charts between two measures $\mu$ and $\nu$ will be indicate with $\mathcal{F}(\mu,\nu)$. We will then define

$$\mathcal{R}(\mu,\nu) = \min_{\mathcal{F}(\mu,\nu)} R(F_1,...,F_d). \tag{20}$$

**Theorem 3.** *Let $\mu$ and $\nu$ be two probability measures over the grid $G = \{1,...,N\}^d$, $c: G \times G \to [0,\infty]$ a separable ground distance, i.e. of the form (??). Then, for each $\pi$ transport plan between $\mu$ and $\nu$ there exists a flow chart $(F_1,..,F_d)$ such that*

$$R(F_1,..,F_d) = \sum_{G \times G} c(a,b)\pi(a,b). \tag{21}$$

*In particular*

$$\mathcal{R}(\mu,\nu) = \mathcal{W}_c(\mu,\nu). \tag{22}$$

*Proof.* Let us consider $\pi$ a transport plan, then we can write

$$\sum_{G \times G} c(a,b)\pi(a,b) = \sum_{a_1,...,a_d,b_1,...,b_d} \sum_{i=1}^{d} \Delta_i(a_i,b_i)\pi(a_1,...,a_d,b_1,...,b_d) =$$

$$\sum_{i=1}^{d} \sum_{a_1,...,a_d,b_1,...,b_d} \Delta_i(a_i,b_i)\pi(a_1,...,a_d,b_1,...,b_d) =$$

$$\sum_{i=1}^{d} \left[ \sum_{b_1,...,b_{i-1},a_i,...,a_d,b_i} \Delta_i(a_i,b_i) f^{(i)}_{b_1,...,b_{i-1},a_i,...,a_d,b_i} \right], \quad (23)$$

where

$$f^{(i)}_{b_1,...,b_{i-1},a_i,...,a_d,b_i} = \sum_{a_1,...,a_{i-1},b_{i+1},...,b_d} \pi(a_1,...,a_d,b_1,...,b_d). \quad (24)$$

To conclude, we have to prove that those $f^{(i)}_{b_1,...,b_{i-1},a_i,...,a_d,b_i}$ satisfy the conditions (**??**), (**??**) and (**??**).

All of those follow from the definition itself, indeed

$$\sum_{b_1} f^{(1)}_{a_1,...,a_d,b_1} = \sum_{b_1,...,b_d} \pi(a_1,...,a_d,b_1,...,b_d) = \mu(a_1,...,a_d),$$

$$\sum_{a_d} f^{(d)}_{b_1,...,b_{d-1},a_d,b_d} = \sum_{a_1,...,a_d} \pi(a_1,...,a_d,b_1,...,b_d) = \nu(b_1,...,b_d)$$

and

$$\sum_{a_i} f^{(i)}_{b_1,...,b_{i-1},a_i,...,a_d,b_i} = \sum_{a_1,...,a_{i-1},a_i,b_{i+1},...,b_d} \pi(a_1,...,a_d,b_1,...,b_d) =$$

$$= \sum_{b_{i+1}} \sum_{a_1,...,a_i,b_{i+2},...,b_N} \pi(a_1,...,a_N,b_1,...,b_N) = \sum_{b_{i+1}} f^{(i+1)}_{b_1,...,b_i,a_{i+1},...,a_N,b_{i+1}}.$$

Let now $(F_1,...,F_d)$ be a flow chart. We have that, for each $i = 1,...,d$, the $F_i$ define a probability measure over $\{1,...,N\}^{d+1}$. For $i = 1$ we easily find that

$$\sum_{a_1,...,a_d,b_1} f^{(1)}_{a_1,...,a_d,b_1} = \sum_{a_1,...,a_d} \sum_{b_1} f^{(1)}_{a_1,...,a_d,b_1} = \sum_{a_1,...,a_d} \mu(a_1,...,a_d) = 1.$$

If we assume that $F_i$ is a probability measure, then, using condition (**??**), we get that

$$\sum_{b_1,...,b_i,a_{i+1},...,a_d,b_{i+1}} f^{(i+1)}_{b_1,...,b_i,a_{i+1},...,a_d,b_{i+1}} = \sum_{b_1,...,b_i,a_{i+1},...,a_d} \sum_{b_{i+1}} f^{(i+1)}_{b_1,...,b_i,a_{i+1},...,a_d,b_{i+1}} =$$

$$\sum_{b_1,...,b_i,a_{i+1},...,a_d} \sum_{a_i} f^{(i)}_{b_1,...,b_{i-1},a_i,...,a_d,b_i} = 1.$$

Thus, by induction, we get that all the $F_i$ are actually probability measures.

Since we showed that $f^{(1)}_{a_1,...,a_d,b_1}$ and $f^{(2)}_{b_1,a_2,...,a_d,b_2}$ are both probability measures and relation (**??**) holds we can apply the gluing lemma and find a probability measure $\pi^{(1)}(a_1,...,a_d,b_1,b_2)$ such that

$$\sum_{b_2} \pi^{(1)}(a_1,...,a_d,b_1,b_2) = f^{(1)}_{a_1,...,a_d,b_1}$$

and

$$\sum_{a_1} \pi^{(1)}_{a_1,...,a_d,b_1,b_2} = f^{(2)}_{b_1,a_2,...,a_d,b_2}.$$

Let us now consider $f^{(3)}_{b_1,b_2,a_3,...,a_d,b_3}$ and $\pi^{(1)}(a_1,...,a_d,b_1,b_2)$, we have

$$\sum_{a_2} \sum_{a_1} \pi^{(1)}(a_1,...,a_d,b_1,b_2) = \sum_{a_2} f^{(2)}_{b_1,a_2,...,a_d,b_2} = \sum_{b_3} f^{(3)}_{b_1,b_2,a_3,...,a_d,b_3},$$

so we can apply once again the gluing lemma and find a probability measure $\pi^{(2)}(a_1, ..., a_d, b_1, b_2, b_3)$ such that
$$\sum_{b_3} \pi^{(2)}(a_1, ..., a_d, b_1, b_2, b_3) = \pi^{(1)}(a_1, ..., a_d, b_1, b_2)$$
and
$$\sum_{a_1, a_2} \pi^{(2)}(a_1, ..., a_d, b_1, b_2, b_3) = f^{(3)}_{b_1, b_2, a_3, ..., a_d, b_3}.$$

We can iterate this process for $d-1$ times and find a probability measure $\pi_{a_1, ..., a_d, b_1, ..., b_d}$ such that

$$\sum_{b_1, ..., b_d} \pi(a_1, ..., a_d, b_1, ..., b_d) = \sum_{b_1, ..., b_{d-1}} \sum_{b_d} \pi(a_1, ..., a_d, b_1, ..., b_d) = \sum_{b_1, ..., b_{N-1}} \pi^{(N-2)}(a_1, ..., a_N, b_1, ..., b_{N-1}) =$$

$$... = \sum_{b_1} \sum_{b_2} \pi^{(1)}(a_1, ..., a_N, b_1, b_2) = \sum_{b_1} f^{(1)}(a_1, ..., a_N, b_1) = \mu(a_1, ..., a_N).$$

Similarly, we have
$$\sum_{a_1, ..., a_d} \pi(a_1, ..., a_d, b_1, ..., b_d) = \nu(b_1, ..., b_d),$$

thus proving that $\pi$ transports $\mu$ into $\nu$.

For such a $\pi$, we now prove that

$$R(F_1, ..., F_d) = \sum_{a_1, ..., a_d, b_1, ..., b_d} \sum_{i=1}^{d} \Delta_i(a_i, b_i) \pi(a_1, ..., a_d, b_1, ..., b_d).$$

We start with

$$\sum_{a_1, ..., a_d, b_1, ..., b_d} \sum_{i=1}^{d} \Delta_i(a_i, b_i) \pi(a_1, ..., a_d, b_1, ..., b_d) = \sum_{i=1}^{d} \sum_{a_1, ..., a_d, b_1, ..., b_d} \Delta_i(a_i, b_i) \pi(a_1, ..., a_d, b_1, ..., b_d).$$

Let us consider the term

$$\sum_{a_1, ..., a_d, b_1, ..., b_d} \Delta_i(a_i, b_i) \pi(a_1, ..., a_d, b_1, ..., b_d) =$$

$$\sum_{b_1, ..., b_{i-1}, a_i, ..., a_d, b_i} \Delta_i(a_i, b_i) \sum_{a_1, ..., a_{i-1}, b_i, ..., b_d} \pi(a_1, ..., a_d, b_1, ..., b_d)$$

but, thanks to the Gluing Lemma, we have that

$$\sum_{a_1, ..., a_{i-1}, b_i, ..., b_d} \pi(a_1, ..., a_d, b_1, ..., b_d) = \sum_{a_1, ..., a_{i-1}, b_i, ..., b_{d-1}} \pi^{(d-2)}(a_1, ..., a_d, b_1, ..., b_{d-1}) = ... =$$

$$\sum_{a_1, ..., a_{i-1}} \pi^{(i+1)}(a_1, ..., a_N, b_1, ..., b_i) = f^{(i)}_{b_1, ..., b_{i-1}, a_i, ..., a_N, b_i}.$$

So the proof is complete. $\qquad\square$