[Reviews · NeurIPS 2018]

Reviewer 1



SUMMARY The authors introduce a new method for solving optimal transport on d-dimensional histograms for separable cost. They exploit this separability to reduce the original linear problem of size n^(2*d) (where n is the number of bin per dimension) to a linear problem of size d*n^(d+1). The resulting algorithm behaves much better than solving the original linear program and becomes de facto the state-of-the-art method for solving exactly optimal transport problem on regular grids. QUALITY Strengths: the paper proposes a new, simple and very efficient way to exploit separability of the cost for solving exactly optimal transport problem. Weakness: apparently, the authors are unaware of "convolutional Wasserstein distances" (see Solomon et al. 2015 and subsequent works) which is the "entropy regularized" counterpart of their approach to *approximately* solve optimal transport in the same setting. There is a nice parallel between these two approaches (factorizing the problem as a sequence of 1-D problems, by only considering "intermediate" marginals of transport plans) that, if mentioned, would further improve the final version of the paper. A comparison to this approach in the numerical section would be fair. CLARITY The article is well written and straight to the point without unnecessary technical details. ORIGINALITY To the best of my knowledge the method is entirely new and is not a simple adaptation of existing methods. SIGNIFICANCE This paper will serve as a reference method for solving optimal transport problems in this setting. COMMENTS - l.46 I'd suggest to end the "previous work" paragraph with a comment that explains what is different in the present contribution. - l.79 do you have a reference for this definition of EMD? Is the denominator used to deal with unbalanced cases? - l.101 this is mainly a matter of taste, but I find that the choice of the indexing variable in Eq.(7) does not help the reader (for instance, I would have opted for mu_(i,j)/nu_(i,j) or mu_(i,a)/nu_(j,b). - l.122 what is a geodetic? did you mean geodesic? In case this is not correct. A mathematical interpretation of the intermediate configuration is as follows: a transport plan pi has 4 indices (a,j,i,b): G1 corresponds to its marginals on (a,j), G2 to its marginals on (i,j) and G3 to its marginals on (i,b). It is a property of separable cost that the total transport cost only depends on these marginals (as you show in Thm.1). - l.186 : Sinkhorn's method can be adapted to separable costs, I suggest to compare with this algorithm for fairness. - l.211 : out of curiosity, why is the bipartite approach so much faster than the EMD code? - l.224 : used *to* study [UPDATE AFTER REBUTTAL: I am still confident that this paper is a good contribution to NIPS. I am glad that the authors chose to include experiments using Convolutional Wasserstein Distances for a fair comparison to "entropy regularized" approaches.]

Reviewer 2



Summary of the paper: This paper presents a fast algorithm for optimal transport, and specifically the 2-Wasserstein distance, that does not rely on regularization. The main contributions of the paper are: 1) an interpretation of the optimal transport problem in d-dimensional space as a network flow over a (d+1)-partite graph; 2) an algorithm that scales very well with the number of grid points and outperforms competing methods. Clarity: Excellent. The paper is very clear, and the algorithm and theoretical contributions were easy to follow. I appreciated the level of detail and illustrative figures. Originality: Above average. Significance: Above average. Detailed comments. Overall, I liked the way the paper is presented. I found the details very clear, and the resulting algorithm is easy to understand and reimplement. My own misgiving is that the method seems restricted to grids for spaces whose ground metric is Euclidean. I have a few extra comments: + In Figure 3, and Section 4 it is not clear how the experiments are structured. What are the source and target distributions for each of the examples? + Table 3 shows some results in higher dimensions, but is still restricted to dimension 4. Is there a small, high-dimensional data set that you could use on which Sinkhorn/LP do not run out of time or memory and on which you could see the dependence of dimension on the running time and memory of your algorithm? + Do you know how/if your method could be adapted to the barycenter problem? [Rebuttal acknowledged; My opinion has not changed while reading the rebuttal. If anything, the authors have responded to all my questions and comments. I still believe the paper is well above the acceptance threshold. The only issue remaining for me is the lack of an example with really high dimensionality.]

Reviewer 3



Update after rebuttal & discussion: I am satisfied with the response by the authors and welcome the addition of further experiments; as I now feel confident the notational and other smaller issues will be fixed adequately, I maintain my score and recommend acceptance of this submission. --------- This paper introduces a new approach to solve optimal transport problems by reformulating the computation of d-dimensional Kantorovich-Wasserstein distances w.r.t. separable cost functions as uncapacitated minimum cost flow problems in (d+1)-partite graphs. The scheme is compared to and contextually embedded into previous works from the literature, and – the main strengths of the paper – the arguments are illustrated by figures and examples, making it easy to follow the contents even if one is only cursorily familiar with optimal transport; numerical experiments are provided to compare the new approach (combined with a network simplex solver for the flow model) with older methods and demonstrate some benefits of the novel formulation. As I found it an interesting read, and somewhat surprising that this simple reformulation has not been thought of before, I believe this work constitutes a worthy contribution to the field and lean toward acceptance of the submission, despite several weaknesses that I shall address point by point below and would like to ask the authors to fix or provide clarification, resp. 1. Please have the paper proof-read for English style and grammar, and typo corrections. 2. I am not sure whether the algorithms you compare to are all still relevant, or if there are better variants that might be more competitive. I got the impression that you are comparing to algorithms that essentially embody the basic principles of one idea or another, but in the Introduction you discuss related works that make it sound like these basic methods have already been improved. Thus, I would appreciate a remark on why you chose the algorithms you did choose in case better/newer variants exist (or if this is not the case and I misread something). 3. Please double-check your notation throughout the paper and consider simplification/clarification where possible. Let me point out some particular issues: 3.a) Why do you introduce X={x_1,...,x_n} (and Y analogously) if it is not really used to simplify/clarify the notation in (1) and (3)-(6) (by using the indices introduced)? For instance, (4) could be written as \sum_{j=1}^{n} \pi(x_i,y_i) \leq \mu_i \forall i=1,\dots,n. Similarly for the Earth Mover's Distance later on. 3.b) Right before the EMD formula on p. 3, what is the notation „ (x_i,\mu(x_i))_i “ (and (y_j,\nu(y_j))_j) to mean exactly (i.e., why the extra subscript i or j, resp.)? Similarly, why the extra subscripts i,j at \mu=\{\mu_{i,j}\}_{i,j} and \nu=... in line 98? Those additional subscripts only clutter the notation and make it less readable, in my opinion; they are not really needed. 3.c) In the definition of \nu (line 55), it should by \nu(y_m) at the end (not \nu(x_m)). 3.d) Direct arcs (a,j) to (i,b) are not present in the model (7)--(10), right? Then why are they introduced in Fig. 2 ? Unless I overlooked something, this does not really make sense and can be quite misleading w.r.t. the tripartite flow model, so please clarify. (I think the direct arcs are from the original bipartite model that the tripartite solutions are mapped back onto via Thm. 1 etc. later... is that so?) 3.e) In Thm. 1, „G \times G“ did not seem right to me; I thought that here, it should be „A^{(1)}\times A^{(2)}“ or „U\times U“? As you can see, I got confused here, and therefore ask to clarify the notation. As a matter of fact, the notation „G“ is overloaded – first, you introduce it to denote an N\times N grid (line 99, p.2) and then to denote the tripartite graph (line 110). Please use distinct letters to avoid mix-ups. (Similarly motivated, I suggest you denote the node subsets in the graph G not by G^{(i)} but rather by V^{(i)}, as the whole node set is called V.) Similarly, in line 147 (p.5), c should probably also be defined on U\times U (or V\times V), not G\times G. (Furthermore, in Thm. 3, G is reused yet again with a different meaning than before, so please also change notation to a distinct letter here to avoid confusion. Also, „separate“ should be „separable“ in Thm. 3.) 3.f) Proof of Thm.1: the second formula is just a term without connection to anything. I suppose it should be equal to the terms in the first formula? In that case, make it a two-line equation; otherwise, please clarify and fix the second formula. 3.g) In Cor.1, it should be „for any discrete measures“ rather than „for each measures“. 3.h) In the proof of Thm. 3 (p.11) that (25), (26) and (27) are pretty much identical to (17), (18) and (19), respectively. These equations are given new numbers here, but later in the proof the earlier numbers are used. So I suggest either not assigning the new numbers (25)-(27) to the known formulas, or else using the new numbers in the proof instead of reverting to the older ones (17)-(19). 4. I was a bit surprised that combinatorial algorithms for the min cost flow problems are not mentioned, but instead, only the network simplex is advocated (for either bi- or (d+1)-partite formulations). Is the network simplex that much better in practice than, say, the classical strongly-polynomial minimum-mean cycle canceling algorithm (A.V. Goldberg & R.E. Tarjan, „Finding minimum-cost circulations by canceling negative cycles“, JACM 36(4), pp.873-886, 1989)? I would appreciate a brief remark on this in the introduction, where the related of optimal transport and min-cost flow problems is described and the network simplex gets its due credit. 5. It should be mentioned that runtimes of (single-thread) C++ code and (multi-thread) Matlab code are not really comparable. Here, this issue is not critical, since the Sinkhorn algorithm (implemented in Matlab) is not an exact method anyway, and the main take-away message is that the novel (d+1)-partite formulation allows significant performance increase compared to the well-known bipartite model. (What language is the EMD code written in? Is it uncomparable to the C++ network simplex, like Sinkhorn, too?) (Also, what is the CPU speed in your test setup, and how many cores are used for the Matlab parallelized code?) 6. In the discussion of the numerical results, you state that „Indeed, the 3-partite formulation is better essentially because it exploits the structure of the ground distance c(x,y) used, that is, the squared $\ell_2$ norm.“ – This is not entirely accurate. True enough, separability allows to reformulate the problem as a min-cost flow problem on a (d+1)-partite graph. The runtime improves significantly then because the runtime of the network simplex (or any min cost flow algorithm) depends, in particular, on the number of arcs in the network. This is what makes the biggest impact in favor of the new formulations, since, e.g., the 3-partite network has more nodes but far fewer edges than the bipartite graph (3N^2 nodes and 2N^3 arcs vs. N^2 nodes and N^4 arcs, i.e., the node number increases only by a constant factor, but the arc number decreases by an order of magnitude, so the difference becomes more severe the larger the networks are)! Please clarify this aspect in the discussion of your experiments. 7. Please rename Section 5, as it does not contain any word of „future work“. (Also, it may want to clarify that the „much smaller flow network“ is smaller in terms of arc number, but does actually have more nodes.) 8. Please fix capitalization issues in the reference list (Sinkhorn, Kantorovich, Wasserstein, Hitchcock, Mallows, Hellinger, Monge. EMD; also capitalize after a colon (:) and in journal titles). Also, please be consistent in how you format the entries (some contain „pages“, others do not; some contain volume and number in the form v(n), others read „volume v“, spell out authors rather than using „et al.“ in the ref. list, and possibly other such things). 9. This may have been a problem with my printer (low ink), but just wondering: Is the suppl. mat. typeset any differently than the paper itself? It is hard to pinpoint, but somehow looks a bit different on my print-outs.